# Bootstrap Approximation of Model Selection Probabilities for Multimodel Inference Frameworks

**DOI:** 10.3390/e26070599

**Published:** 2024-07-15

**Authors:** Andres Dajles, Joseph Cavanaugh

**Affiliations:** Department of Biostatistics, University of Iowa, 145 N. Riverside Drive, Iowa City, IA 52242, USA; joe-cavanaugh@uiowa.edu

**Keywords:** Akaike information criterion, Akaike weights, Bayesian model averaging, bootstrapping, model selection

## Abstract

Most statistical modeling applications involve the consideration of a candidate collection of models based on various sets of explanatory variables. The candidate models may also differ in terms of the structural formulations for the systematic component and the posited probability distributions for the random component. A common practice is to use an information criterion to select a model from the collection that provides an optimal balance between fidelity to the data and parsimony. The analyst then typically proceeds as if the chosen model was the only model ever considered. However, such a practice fails to account for the variability inherent in the model selection process, which can lead to inappropriate inferential results and conclusions. In recent years, inferential methods have been proposed for multimodel frameworks that attempt to provide an appropriate accounting of modeling uncertainty. In the frequentist paradigm, such methods should ideally involve model selection probabilities, i.e., the relative frequencies of selection for each candidate model based on repeated sampling. Model selection probabilities can be conveniently approximated through bootstrapping. When the Akaike information criterion is employed, Akaike weights are also commonly used as a surrogate for selection probabilities. In this work, we show that the conventional bootstrap approach for approximating model selection probabilities is impacted by bias. We propose a simple correction to adjust for this bias. We also argue that Akaike weights do not provide adequate approximations for selection probabilities, although they do provide a crude gauge of model plausibility.

## 1. Introduction

When presented with a set of variables posited as potential predictors of a targeted outcome, researchers harness an array of techniques to construct an appropriate descriptive or predictive model. Historically, the conventional approach entailed the formulation of a model under the guidance of an expert, who, grounded in a scientific understanding of the underlying mechanics of the observed outcome, would advocate for the structure of the proposed model. In contemporary practice, while domain expertise continues to inform the compilation of potential explanatory variables, the adoption of an all-encompassing model incorporating all of these variables is rare. Instead, scientists and statisticians have gravitated towards model selection algorithms (e.g., best subsets selection, forward selection, backward elimination, stepwise selection, and the LASSO), which utilize the sample at hand to yield models exhibiting more parsimonious structures than the comprehensive global model.

Once a model is selected, the common practice is to conduct inference on that model, proceeding as if this were the only model to ever be considered. However, different samples from the same population may lead to a selection of models with different structures. Therefore, the standard practice neglects the sampling variability inherent in the selection process. This pervasive issue in contemporary applications of statistics was described by Breiman as a “quiet scandal in the statistical community” [1].

A direct consequence of this oversight is the inherent difficulty in replicating the results of a modeling analysis using subsequent samples [2,3]. Therefore, inferences contingent on the chosen model are profoundly impacted, raising concerns regarding the bias of regression effect estimators, the accuracy of their estimated standard errors, and the validity of associated *p*-values or confidence intervals [3]. For instance, failing to account for model selection variability will typically result in smaller standard errors, which in turn results in erroneously smaller *p*-values and overly optimistic, narrower confidence intervals.

In addition, common model selection procedures are prone to including spurious effects in the final model [4,5]. In the setting of regression models with correlated variables, the inclusion of spurious effects can heavily influence the estimates and the interpretation of important effects. For example, if two explanatory variables are correlated, their estimated effects can vastly vary for models that only include one of the variables as opposed to models that include both variables [3].

In recent years, various approaches have been proposed that are aimed at addressing the estimation of regression effects while accounting for model selection variability. These approaches can often be characterized as multimodel inference procedures, where regression effect estimates are not solely derived from a single model but rather from the entire collection of models. Conceptually, such procedures are aligned with ensemble methods, where the final prediction or classification is derived through the combination of multiple submodels.

The foundational principles of multimodel inference are influenced by Bayesian principles, particularly the concept of Bayesian model averaging (BMA). The BMA framework is founded on the recognition that different estimates are contingent upon specific models, and that these models possess varying probabilities of arising, given the observed data. Consequently, the posterior mean of any quantity of interest should encompass contributions from all the models based on their posterior probabilities. In essence, the posterior mean of the target quantity should be an average of the estimates conditional on each model, weighted by the posterior probability of that particular model.

In the frequentist domain, several multimodel inference methodologies mirror the BMA framework. A prominent approach involves the utilization of Akaike weights to play the role of model probabilities [6]. These Akaike weights offer a measure of evidence for each model in a candidate set, allowing us to compute the estimate of each regression coefficient by aggregating their Akaike-weighted model-specific estimates. In this manner, a model averaging calculation akin to BMA is performed but with the incorporation of each model’s Akaike weight as opposed to its posterior probability.

The basis of incorporating Akaike weights into the multimodel inference framework presupposes the validity of Akaike weights as approximations to model selection probabilities. In fact, Burnham and Anderson claim that Akaike weights “may be interpreted as the probability that model *i* is the actual expected K–L [Kullback–Leibler] best model for the sampling situation considered” [6]. However, in this paper, we argue that Akaike weights do not provide adequate approximations to model probabilities. Instead, we demonstrate that repeating the model selection process via the bootstrap yields better approximations to model selection probabilities, which can then be incorporated for the purpose of conducting multimodel inference. Yet this bootstrap procedure must be implemented with caution because of the bias generated from bootstrapping the likelihood-based information criteria. Under appropriate conditions, we derive the form of this bias and propose a simple correction.

## 2. Akaike Weights

### 2.1. Background

The Akaike information criterion (AIC), introduced by Hirotugu Akaike in his seminal 1973 publication “Information Theory and an Extension of the Maximum Likelihood Principle”, has emerged as a pivotal tool in statistical model selection. To appreciate the evolution of the AIC framework, it is important to recall that the likelihood function encapsulates the fidelity of a model to the observed data. Consequently, it serves as a potent instrument for discerning the propriety of different candidate models. However, the adherence of the likelihood function to the data increases in tandem with the complexity of the model. If models are assessed based on the magnitude of the empirical likelihood, higher-dimensional models will invariably be favored, thus potentially culminating in the selection of models with excessively complex structures. Therefore, to employ the maximum likelihood framework for model selection, a recalibration of the likelihood function is necessary so that the favorability of complexity is conferred solely when warranted. This essential adaptation is achieved by AIC.

For any prospective model parameterized by θ, the AIC is given by
(1)$$-2\ell (\hat{\theta }|y)+2k,$$
where *k* denotes the dimension of θ, and $\hat{\theta }$ denotes the maximum likelihood estimate of the parameter vector θ. The statistic $-2\ell (\hat{\theta }|y)$ is known as the “goodness-of-fit” term, and represents the log of the empirical likelihood for the model parameterized by θ based on the data *y*. Ergo, this term inherently gravitates towards models exhibiting intricate configurations, thus conceivably engendering a predisposition towards complexity. Counterbalancing this predilection, the term 2k is known as the “penalty”, which increases as the model complexity escalates. In the pursuit of selecting an optimal model from a candidate collection, the objective is to choose the model exhibiting the minimal AIC value. This AIC value represents an equilibrium between the “goodness-of-fit” term and the “penalty” term. To elucidate this point, a model of pronounced intricacy is able to assimilate the subtleties inherent in the data, consequently resulting in a diminished “goodness-of-fit” term. However, the associated complexity results in a high “penalty” term, which means the more complex models may not necessarily produce the smallest AIC value.

In the traditional linear regression setting, consider a vector of outcomes *y* that we aim to model using a set of (p−1) explanatory variables plus an intercept term, represented by the rank-*p* design matrix *X*. Let $\theta ={\left[\begin{array}{cc} \beta^{T} & \sigma^{2} \end{array}\right]}^{T}$, where $\beta^{T}=\left[\begin{array}{ccc} \beta_{1} & \ldots & \beta_{p} \end{array}\right]$ denotes the corresponding regression coefficients, and let $M=\{M_{1},M_{2},\ldots ,M_{i},\ldots ,M_{2^{\left(p-1\right)}}\}$ represent the set of all conceivable candidate models that can be constructed from the (p−1) variables. Additionally, let $AIC_{i}$ denote the value of AIC associated with the $i^{th}$ model $M_{i}\in M$, and let $AIC_{min}$ denote the minimum AIC value across all the models in the candidate set *M*. In other words, we have that
$$AIC_{min}=min\left(\{AIC_{1},AIC_{2},\ldots ,AIC_{2^{\left(p-1\right)}}\}\right).$$ If we then let $\Delta_{i}=AIC_{i}-AIC_{min}$, the Akaike weight for model $M_{i}$ is defined as
(2)$$w_{i}=\frac{e^{-0.5\Delta_{i}}}{\sum_{j=1}^{2^{\left(p-1\right)}}e^{-0.5\Delta_{j}}}.$$

### 2.2. Akaike Weights vs. Model Probabilities

Akaike weights have been frequently utilized as approximations to model selection probabilities. At first glance, this seems like a reasonable practice. For instance, Equation (Equation 2) has a similar structure to the Bayesian posterior model probability approximated via the Bayesian information criterion (BIC); see, for instance, the work of Neath and Cavanaugh [7]. However, although Burnham and Anderson advocate for the use of Akaike weights as approximations for model selection probabilities, they also recognize that such an approximation may be crude [6]. Through the following example, we argue that Akaike weights do not serve as adequate approximations to selection probabilities.

Consider a simple case where we have a null and an alternative model represented by $M_{0}$ and $M_{1}$, respectively. Then, the corresponding Akaike weights are given by
$$w_{0}=\frac{e^{-0.5\Delta_{0}}}{e^{-0.5\Delta_{0}}+e^{-0.5\Delta_{1}}}$$
and
$$w_{1}=\frac{e^{-0.5\Delta_{1}}}{e^{-0.5\Delta_{0}}+e^{-0.5\Delta_{1}}}.$$ However, the Akaike weights can be rewritten as follows:$$\begin{array}{cc} w_{1} & =\frac{e^{-0.5\Delta_{1}}}{e^{-0.5\Delta_{0}}+e^{-0.5\Delta_{1}}}\\ \; & =\frac{\frac{e^{-0.5\Delta_{1}}}{e^{-0.5\Delta_{1}}}}{\frac{e^{-0.5\Delta_{0}}+e^{-0.5\Delta_{1}}}{e^{-0.5\Delta_{1}}}}\\ \; & =\frac{1}{1+e^{-0.5\left(AIC_{0}-AIC_{min}\right)+0.5\left(AIC_{1}-AIC_{min}\right)}}\\ \; & =\frac{1}{1+e^{-0.5(AIC_{0}-AIC_{1})}}. \end{array}$$ Similarly,
$$w_{0}=\frac{1}{1+e^{\;0.5(AIC_{0}-AIC_{1})}}.$$

Now, note the following:$$\begin{array}{cc} \; & w_{1}=\frac{1}{1+e^{-0.5(AIC_{0}-AIC_{1})}}\Rightarrow \\ \; & w_{1}+w_{1}e^{-0.5(AIC_{0}-AIC_{1})}=1\Rightarrow \\ \; & w_{1}e^{-0.5(AIC_{0}-AIC_{1})}=1-w_{1}\Rightarrow \\ \; & w_{0}=w_{1}e^{-0.5(AIC_{0}-AIC_{1})}\Rightarrow \;\mathrm{because}\;w_{0}=1-w_{1}\\ \; & \frac{w_{0}}{w_{1}}=e^{-0.5(AIC_{0}-AIC_{1})}\Rightarrow \\ \; & \log\left(\frac{w_{0}}{w_{1}}\right)=-0.5\left(AIC_{0}-AIC_{1}\right). \end{array}$$

This derivation implies that if the Akaike weights are treated as actual model choice probabilities, then for models $M_{i}$ and $M_{j}$ such that i≠j, the log of the corresponding relative risk is proportional to $AIC_{i}-AIC_{j}$. However, this property is erroneous, in that the relative risk based on the actual model selection probabilities will not satisfy this proportionality.

To illustrate this incongruity, consider the case where $M_{0}$ and $M_{1}$ are parameterized by $\theta_{0}$ and $\theta_{1}$, respectively, but $|\theta_{1}\left|-\right|\theta_{0}|=1$; in other words, the dimension of the parameter vectors only differs by one. In addition, assume that the null is the true data-generating model. Note that in this case, $\Delta_{0}=0\Rightarrow AIC_{0}-AIC_{1}\leq 0$; therefore, if we let $P(M_{0})$ be the selection probability for model $M_{0}$, then $P(M_{0})$ is equivalent to $P(AIC_{0}-AIC_{1}\leq 0).$ Under a true null model, we can write
$$\begin{array}{cc} P(M_{0}) & =P(AIC_{0}-AIC_{1}\leq 0)\\ \; & =P(-2\left(\ell \left({\hat{\theta }}_{0}|y\right)-\ell \left({\hat{\theta }}_{1}|y\right)\right)-2\leq 0)\\ \; & \approx P(\chi_{1}^{2}\leq 2)\\ \; & =0.8427 \end{array}$$ This means that asymptotically, the relative risk of $M_{0}$ over $M_{1}$ is equal to
$$\begin{array}{cc} \frac{P(M_{0})}{P(M_{1})} & =\frac{0.8427}{1-0.8427}\\ \; & =5.3573. \end{array}$$

For the sake of argument, suppose that $w_{0}$ and $w_{1}$ can serve as approximations to $P(M_{0})$ and $P(M_{1})$. However, note that
$$\begin{array}{cc} \frac{w_{0}}{w_{1}} & =e^{-0.5(AIC_{0}-AIC_{1})}\\ \; & <e^{-0.5(-2)}\;\mathrm{because}-2(\ell \left({\hat{\theta }}_{0}|y\right)-\ell \left({\hat{\theta }}_{1}|y\right))>0\;\\ \; & =e^{1}\\ \; & =2.7183. \end{array}$$

Since the ratio $\left(w_{0}/w_{1}\right)$ can never attain the actual value of the relative risk, the Akaike weights should not be treated as approximations to the probability of choosing a particular model.

## 3. Bootstrap Model Frequencies

### 3.1. Model Frequencies as Multinomial Probability Vector Approximations

Consider a scenario in which we can draw a set of *S* samples from the true data-generating model, denoted by *g*. For each sample s∈S, we evaluate the quantity $\Delta_{i}=AIC_{i}-AIC_{min}$.

Now, it is worth noting that if $M_{i}$ emerges as the optimal model for sample *s*, then $AIC_{i}=AIC_{min}$, which implies that $\Delta_{i}=0$ for such a sample. Subsequently, we define a random vector $\tau^{T}=\left(\tau_{1},\tau_{2},\ldots ,\tau_{2^{\left(p-1\right)}}\right)$ to denote the number of selections for each model across all *S* draws. Therefore,
$$\tau_{i}=\sum_{s=1}^{S}I\left(\Delta_{i}\left(s\right)=0\right),$$
where $\Delta_{i}\left(s\right)$ denotes the AIC difference Δ(i) for sample *s*.

Given this configuration, we can assert that τ∼multinomial(S,π), where *S* represents the number of trials, and $\pi^{T}=\left(\pi_{1},\pi_{2},\ldots ,\pi_{2^{\left(p-1\right)}}\right)$ constitutes a vector of probabilities. Specifically, $\pi_{i}$ corresponds to $P(M_{i})$, the probability of selecting model $M_{i}$. Furthermore, since model $M_{i}$ is exclusively chosen when $\Delta_{i}=0$, we deduce that $\pi_{i}=P\left(\Delta_{i}=0\right)$. Finally, we obtain:$$\begin{array}{cc} \pi^{T} & =E_{g}\left({\hat{\pi }}^{T}\right)\\ \; & =E_{g}\left(\frac{\sum_{s=1}^{S}I\left(\Delta_{1}\left(s\right)=0\right)}{S},\ldots ,\frac{\sum_{s=1}^{S}I\left(\Delta_{2^{\left(p-1\right)}}\left(s\right)=0\right)}{S}\right). \end{array}$$

Thus, we can define model selection probabilities as the probability vector π from a multinomial distribution with *S* number of trials, where each trial s∈S is a sample from the true data generating model. Therefore, a model selection probability approximation should be an attempt to estimate π. For this task, we will employ the non-parametric bootstrap, though we need to consider a set of important caveats delineated in Section 3.2.

### 3.2. Bias Induced from Bootstrapping Fitted Likelihoods

Let $y^{T}=\left(y_{1},\ldots ,y_{n}\right)$ denote the observation vector, which we assume is comprised of *n* independent variates $y_{i}$. To assess the disparity between a data generating model g(y) and a parametric approximating model f(y|θ), we can employ the Kullback–Leibler information between g(y) and f(y|θ) with respect to g(y), given by
$$I_{KL}\left(g,\theta \right)=E_{g}\left[\log\frac{g(y)}{f(y|\theta )}\right].$$

The expression
$$d\left(g,\theta \right)=E_{g}[-2\ell \left(\theta |y\right)],$$
known as the Kullback–Leibler discrepancy (KLD), is often used as a substitute for $I_{KL}$ since $E_{g}\left[\log g\left(y\right)\right]$ does not depend on the structure of the approximating model f(y|θ).

In practice, the goal is to determine the propriety of fitted models of the form $f(y|\hat{\theta })$, where $\hat{\theta }={argmax}_{\theta \in \Theta }\;\ell (\theta |y)$. The KLD for the fitted model is given by
(3)$$d\left(g,\hat{\theta }\right)=E_{g}\left[-2\ell \left(\theta |y\right)\right]{|}_{\theta =\hat{\theta }}.$$

To develop bootstrap approximations to model selection probabilities, we will use the “plug-in” principle as described by Efron and Tibshirani [8]. In our current application, we replace the true data generating distribution *g* by the empirical distribution $\hat{g}$; the original sample *y* by a bootstrap sample $y^{*}$ drawn from $\hat{g}$; and finally, $\hat{\theta }$ by the maximum likelihood estimate (MLE) ${\hat{\theta }}^{*}$ derived under the bootstrap sample $y^{*}$. For clarification purposes, it is worth emphasizing that $E_{g}\left(\cdot \right)$ is the expected value with respect to the data generating distribution, while $E_{*}\left(\cdot \right)$ is the expected value with respect to the bootstrap distribution. In fact, any notation presented with the symbol * will correspond to a bootstrap-based construct. For example, the bootstrap version of AIC is defined as
(4)$$AIC^{*}=-2\ell \left({\hat{\theta }}^{*}|y^{*}\right)+2k.$$

With the preceding replacements, the bootstrap version of the KLD is given by
$$\begin{array}{cc} d(\hat{g},{\hat{\theta }}^{*}) & =E_{\hat{g}}{\left[-2\ell \left(\theta |y\right)\right]|}_{\theta ={\hat{\theta }}^{*}}\\ \; & =\sum_{i=1}^{N}-2\ell_{i}\left({\hat{\theta }}^{*}|y_{i}\right)\;(\mathrm{because}\;\mathrm{each}\;y_{i}\;\mathrm{is}\;\mathrm{independent}.)\\ \; & =-2\ell ({\hat{\theta }}^{*}|y), \end{array}$$
where $\ell_{i}$ represents the contribution to the likelihood based on the *i*th response $y_{i}$.

The work by Dajles and Cavanaugh [9] shows that
(5)$$E_{g}\left\{E_{*}\left[-2\ell \left({\hat{\theta }}^{*}|y\right)\right]\right\}\approx E_{g}\left[d\left(g,\hat{\theta }\right)\right]-k$$
where *k* is the dimension of the model. With these results, we can provide the following lemma.

**Lemma 1.** 
*Consider a large-sample setting and assume that the candidate model parameterized by θ subsumes the true model. Then, for the corresponding empirical log-likelihood given by $\ell (\hat{\theta }|y)$, we have that*

$$E_{g}\left[-2\ell (\hat{\theta }|y)\right]\approx E_{g}\left[E_{*}\left[-2\ell ({\hat{\theta }}^{*}|y^{*})\right]\right]+k$$

*where k=|θ| and $E_{*}$ is the expectation with respect to the bootstrap distribution.*


**Proof.** Let $d(g,\hat{\theta })$ as defined in (Equation 3) denote the KLD for the fitted candidate model.Now, consider the extended information criterion (EIC) [10] for this fitted model, which is given by
$$EIC=-2\ell \left(\hat{\theta }|y\right)+2E_{*}\left(\ell \left({\hat{\theta }}^{*}|y^{*}\right)-\ell \left({\hat{\theta }}^{*}|y\right)\right).$$Under the assumption that the candidate model subsumes the true model, it has been shown that EIC is an asymptotically unbiased estimator of the expected KLD, which means that $E_{g}\left(d\left(g,\hat{\theta }\right)\right)\approx E_{g}\left(EIC\right)$ [10]. Therefore, by applying the expected value under the generating model *g* to the expression for EIC, we obtain the following:
$$\begin{array}{cc} E_{g}\left(d\left(g,\hat{\theta }\right)\right) & \approx E_{g}\left[-2\ell \left(\hat{\theta }|y\right)+2E_{*}\left(\ell \left({\hat{\theta }}^{*}|y^{*}\right)-\ell \left({\hat{\theta }}^{*}|y\right)\right)\right]\\ \; & =E_{g}\left[-2\ell (\hat{\theta }|y)+2k\right]-2k+E_{g}\left[2E_{*}\left(\ell \left({\hat{\theta }}^{*}|y^{*}\right)-\ell \left({\hat{\theta }}^{*}|y\right)\right)\right]\;\\ \; & =E_{g}\left[AIC\right]+E_{g}\left[E_{*}\left(2\ell ({\hat{\theta }}^{*}|y^{*})-2k\right)\right]-E_{g}\left[E_{*}\left(2\ell ({\hat{\theta }}^{*}|y)\right)\right]\;\mathrm{by}\;(1)\\ \; & =E_{g}\left[AIC\right]-E_{g}\left[E_{*}\left(-2\ell ({\hat{\theta }}^{*}|y^{*})+2k\right)\right]-E_{g}\left[E_{*}\left(2\ell ({\hat{\theta }}^{*}|y)\right)\right]\;\\ \; & =E_{g}\left[AIC\right]-E_{g}\left[E_{*}\left(AIC^{*}\right)\right]-E_{g}\left[E_{*}\left(2\ell ({\hat{\theta }}^{*}|y)\right)\right]\mathrm{by}\;(4). \end{array}$$Now, under the same modeling specification assumption, we know that AIC is also an asymptotically unbiased estimator of the expected KLD. Therefore,
$$\begin{array}{cc} \; & E_{g}\left(d\left(g,\hat{\theta }\right)\right)\approx E_{g}\left[AIC\right]-E_{g}\left[E_{*}\left(AIC^{*}\right)\right]-E_{g}\left[E_{*}\left(2\ell ({\hat{\theta }}^{*}|y)\right)\right]\Rightarrow \\ \; & 0\approx -E_{g}\left[E_{*}\left(AIC^{*}\right)\right]-E_{g}\left[E_{*}\left(2\ell ({\hat{\theta }}^{*}|y)\right)\right]\Rightarrow \\ \; & E_{g}\left[E_{*}\left(AIC^{*}\right)\right]\approx E_{g}\left[E_{*}\left(-2\ell ({\hat{\theta }}^{*}|y)\right)\right]. \end{array}$$Recall the approximation (Equation 5), which states that
$$E_{*}\left(-2\ell ({\hat{\theta }}^{*}|y)\right)\approx AIC-k.$$Hence, we have that
$$\begin{array}{cc} E_{g}\left[E_{*}\left(AIC^{*}\right)\right] & \approx E_{g}\left[E_{*}\left(-2\ell ({\hat{\theta }}^{*}|y)\right)\right]\Rightarrow \\ E_{g}\left[E_{*}\left(AIC^{*}\right)\right] & \approx E_{g}\left[AIC-k\right]\;\mathrm{by}\;\left(5\right)\Rightarrow \\ E_{g}\left[AIC\right] & \approx E_{g}\left[E_{*}\left(AIC^{*}\right)\right]+k\Rightarrow \\ E_{g}\left[-2\ell (\hat{\theta }|y)\right] & \approx E_{g}\left[E_{*}\left(-2\ell ({\hat{\theta }}^{*}|y^{*})\right)\right]+k. \end{array}$$The preceding establishes the lemma. □

The results from Lemma 1 indicate that if we want to build a bootstrap distribution of AIC variates for which the expected value of each variate is approximately unbiased for the expected value of the fitted KLD, then we must adjust the variates by *k*. Moreover, the results from Lemma 1 in combination with the expression (Equation 5) reveal a deeper understanding of the behavior of fitted log-likelihoods when subjected to a bootstrap procedure.

For the purpose of presenting parallel notation, employing the trivial result that $-2\ell \left(\hat{\theta }|y\right)=E_{*}\left[-2\ell (\hat{\theta }|y)\right]$, we have the following asymptotic results: (6)$$\begin{array}{cc} E_{g}\left[d(g,\hat{\theta })\right] & \approx E_{g}\left\{E_{*}\left[-2\ell \left({\hat{\theta }}^{*}|y\right)\right]\right\}+k\;\mathrm{by}\;\mathrm{Equation}\;\left(5\right) \end{array}$$(7)$$\begin{array}{cc} E_{g}\left[d(g,\hat{\theta })\right] & \approx E_{g}\left\{E_{*}\left[-2\ell \left(\hat{\theta }|y\right)\right]\right\}+2k\;\mathrm{by}\;\mathrm{the}\;\mathrm{properties}\;\mathrm{of}\;\mathrm{AIC} \end{array}$$(8)$$\begin{array}{cc} E_{g}\left[d(g,\hat{\theta })\right] & \approx E_{g}\left\{E_{*}\left[-2\ell \left({\hat{\theta }}^{*}|y^{*}\right)\right]\right\}+3k\;\mathrm{by}\;\mathrm{Lemma}\;\left(1\right) \end{array}$$

Equations (Equation 6)–(Equation 8) exhibit the intricate connection between the penalty term and the level of optimism conveyed by fitted log-likelihoods in the context of the bootstrap. In Equation (Equation 6), the estimate of the parameter θ is based on the bootstrap data, and the likelihood evaluates the efficacy of the fitted model parameterized by ${\hat{\theta }}^{*}$ in predicting the original sample *y*. This corresponds to the classical setting in which the bootstrap is used to assess the predictive accuracy of a fitted model. Since some of the elements of *y* will not be represented in $y^{*}$, one can think of this setting as representing pseudo out-of-sample prediction. Here, the penalty term should be *k*, the number of parameters in the model.

Now, when we look at Equation (Equation 7), the likelihood evaluates the efficacy of the fitted model parameterized by $\hat{\theta }$ in predicting the sample *y* that was used to obtain ${\hat{\theta }}^{*}$. The strong dependence of $\hat{\theta }$ and *y* results in overly optimistic prediction, thus justifying the need to increase the penalty to 2k. Analogously, when we look at Equation (Equation 8), the likelihood evaluates the efficacy of the fitted model parameterized by the bootstrap replicate ${\hat{\theta }}^{*}$ in predicting the bootstrap sample $y^{*}$ used to obtain ${\hat{\theta }}^{*}$. The strong dependence of ${\hat{\theta }}^{*}$ and $y^{*}$ again results in overly optimistic prediction. Moreover, the optimism is increased by the duplicated elements in the fitting sample $y^{*}$, i.e., there are more independent pieces of information in *y* than in $y^{*}$. Here, the penalization needs to be increased to 3k.

### 3.3. Bootstrap Approximation of Model Selection Probabilities

In Section 3.1, we introduced the idea of the model selection probability as an entry of a probability vector in a multinomial distribution. With this framework, we have that
$$\begin{array}{cc} \pi^{T} & =E_{g}\left({\hat{\pi }}^{T}\right)\\ \; & =E_{g}\left(\frac{\sum_{s=1}^{S}I\left(\Delta_{1}\left(s\right)=0\right)}{S},\ldots ,\frac{\sum_{s=1}^{S}I\left(\Delta_{2^{\left(p-1\right)}}\left(s\right)=0\right)}{S}\right)\\ \; & \approx E_{*}\left(\frac{\sum_{b=1}^{B}I\left(\Delta_{1}^{*}\left(b\right)=0\right)}{B},\ldots ,\frac{\sum_{b=1}^{B}I\left(\Delta_{2^{\left(p-1\right)}}^{*}\left(b\right)=0\right)}{B}\right), \end{array}$$
where $E_{*}$ is the expected value with respect to the bootstrap distribution. Here, $\Delta_{i}^{*}=AIC_{i,adj}^{*}-AIC_{min,adj}^{*}$, with $AIC_{i,adj}$ denoting the bias-adjusted bootstrap variant of AIC for model $M_{i}$: $AIC_{i,adj}^{*}=AIC_{i}^{*}+k_{i}$, $k_{i}=\left|\theta_{i}\right|$, and with
$$AIC_{min,adj}^{*}=\min\left(\left\{AIC_{1,adj}^{*},\ldots ,AIC_{2^{\left(p-1\right)},adj}^{*}\right\}\right).$$

Thus, for model $M_{i}$, we define the Bootstrap Model Frequency (BMF) by
$$P^{*}\left(M_{i}\right)=\frac{\sum_{b=1}^{B}I\left(\Delta_{i}^{*}\left(b\right)=0\right)}{B},$$
which serves as a bootstrap approximation of $P(M_{i})$.

### 3.4. Akaike Weights vs. Bootstrap Model Frequencies: Simulation

To better understand the behavior of Akaike weights vs. BMFs, consider the true data-generating model given by
$$y_{i}=\beta_{0,1}+\epsilon_{i},$$
with $\epsilon_{i}∼N\left(0,1\right)$ and $\beta_{0,1}=1$. Now, suppose the null and alternative candidate models under consideration have the following form:$$\begin{array}{cc} \; & M_{0}:y_{i}=\beta_{1}+\eta_{i},\\ \; & M_{1}:y_{i}=\beta_{1}+\beta_{2}x_{i2}+\eta_{i}, \end{array}$$
where $x_{i2}∼N\left(0,1\right)$ and $\eta_{i}∼N\left(0,\sigma^{2}\right)$. Thus, the null model is correctly specified and the alternative is overspecified. For 1000 bootstrap iterations and 500 samples from the data-generating model, each of size N=20, we can compute the Akaike weights and the AIC-based bootstrap model frequencies for $M_{0}$ and $M_{1}$. The results are displayed in Figure 1. Here, the discrepancy between Akaike weights and BMFs is noticeable.

The red, vertical dotted lines in Figure 1 represent the boundaries associated with the Akaike weights. More precisely, similarly to the log of the relative risk, the Akaike weights are bounded by a function of the minimum difference between the empirical log-likelihoods of the null and the alternative models. In other words, since $-2(\ell \left({\hat{\theta }}_{0}|y\right)-\ell \left({\hat{\theta }}_{1}|y\right))>0$, then
$$\begin{array}{cc} w_{0} & =\frac{1}{1+e^{\;0.5(AIC_{0}-AIC_{1})}}\\ \; & <\frac{1}{1+e^{\;0.5(-2)}}\\ \; & =0.7311, \end{array}$$
and similarly,
$$\begin{array}{cc} w_{1} & =\frac{1}{1+e^{-0.5(AIC_{0}-AIC_{1})}}\\ \; & >\frac{1}{1+e^{-0.5(-2)}}\\ \; & =0.2689. \end{array}$$

Table 1 shows the average of the Akaike weights and the BMFs over all 500 simulated data sets. Herein, we see that the average Akaike weights and the average BMFs are not close to each other. However, although not identical in value, the average BMF is close to the empirical estimate of the true model probability. For these 500 data sets, we have that $P(M_{0}=0)\approx 0.8440$ and
$$\sum_{s=1}^{500}\frac{P^{*}\left(\Delta_{0}^{*}\left(s\right)=0\right)}{500}\approx 0.7566.$$ Conversely, the average Akaike weight for $M_{0}$ is 0.6166, which is far from the empirical estimate of $P(M_{0})$.

### 3.5. More Complex Simulation Scenarios

The setting explored in Section 3.4 provides an intuitive and illustrative example of the behavior of Akaike weights and BMFs and the differences between them. However, one can consider other more complex simulation scenarios, though perhaps at the expense of mathematical clarity.

For instance, consider the data generating model given by
$$y_{i}=1+2x_{i1}+1x_{i2}+0x_{i3}+\epsilon_{i},$$
with $\epsilon_{i}∼N\left(0,1\right)$, and
$${\left[\begin{array}{ccc} x_{i1} & x_{i2} & x_{i3} \end{array}\right]}^{T}∼N_{3}\left(\mu ,\Sigma \right),$$
where μ={1,1,1} and $\Sigma =\left[\begin{array}{ccc} 1 & 0.5 & 0.5\\ 0.5 & 1 & 0.5\\ 0.5 & 0.5 & 1 \end{array}\right].$

In this setting, of the eight possible candidate models, only two models subsume the correct mean structure, while the remaining models are underspecified. Moreover, to test the versatility of BMFs, we also introduce some correlation between the explanatory variables.

Table 2 shows the performance one would expect given the theoretical underpinnings that we have developed and presented. In approximating the model probabilities for the small sample size, the performance of the average BMF is not ideal but is still superior to that of the Akaike weights. However, as the sample size increases, we observe a significant improvement in BMF performance. For the sample size of N=1000, we note that the average BMFs are sufficiently close to their respective model selection probabilities.

Now, consider a more complex case with the data generating model given by
$$y_{i}=1+2x_{i1}+1x_{i2}+0x_{i3}+\epsilon_{i},$$
with $\epsilon_{i}∼Z\cdot N\left(0,3\right)+\left(1-Z\right)\cdot N\left(0,1\right)$, Z∼Bernoulli(π), π=0.8, and
$${\left[\begin{array}{ccc} x_{i1} & x_{i2} & x_{i3} \end{array}\right]}^{T}∼N_{3}\left(\mu ,\Sigma \right),$$
where μ={1,1,1} and $\Sigma =\left[\begin{array}{ccc} 1 & 0.5 & 0.5\\ 0.5 & 1 & 0.5\\ 0.5 & 0.5 & 1 \end{array}\right].$

In this setting, assuming that the models are fit using a Gaussian log-likelihood, each candidate model is misspecified with respect to the error distribution. With regard to the mean structure, two models subsume the correct structure, while the remaining models are underspecified. Table 3 illustrates that for a sufficiently large sample size (N=1000), the adjusted bootstrap model frequencies (BMFs) outperform the Akaike weights. As the sample size decreases, we observe that the BMFs diverge from the simulated model selection probabilities.

The performance observed in the large sample size scenario aligns with theoretical expectations. For any information criterion based on empirical log-likelihoods, if the sample size is sufficiently large relative to the number of variables in the candidate models, the goodness-of-fit term will predominate and favor the selection of larger models that subsume the appropriate mean structure. Since the true model selection probabilities are calculated via AIC, model misspecification is likely to similarly affect both the BMFs and the simulated true model probabilities. However, keep in mind that for this setting, AIC is not guaranteed to provide an asymptotically unbiased estimator of the KLD, which means that the simulated true model selection probabilities will not be correctly calibrated.

As the sample size decreases, the penalty term becomes more influential, and the effects of violations to the regularity conditions under which AIC is asymptotically justified become more pronounced. Therefore, in small sample size settings, we expect to see greater discrepancies in model selection between the simulated samples and the bootstrap samples. Precisely elucidating these discrepancies is challenging and would require a rigorous mathematical exploration of the effect of misspecification on the performance of AIC as an asymptotically unbiased estimator of the expected KLD. To avoid mathematical ambiguities, it is best to consider a simple case as demonstrated in Section 3.4.

### 3.6. Bootstrap-Based Multimodel Estimates and Confidence Intervals

Bootstrap model frequencies are an important component of the overall multimodel inference framework. Procedures for obtaining bootstrap-based estimates and confidence intervals within this framework are rigorously outlined in Efron’s seminal work [11]. In our current contribution, we apply the concept of “bootstrap smoothing” to the estimation of regression coefficients. The algorithm can be summarized as follows.

Consider a vector of outcomes *y* that we aim to model using a set of (p−1) explanatory variables plus an intercept term, represented by the design matrix *X*. Let $\beta^{T}=\left[\begin{array}{ccc} \beta_{1} & \ldots & \beta_{p} \end{array}\right]$ denote the corresponding coefficients, and let
$$M=\{M_{1},M_{2},\ldots ,M_{i},\ldots ,M_{2^{\left(p-1\right)}}\}$$
represent the set of all conceivable candidate models that can be constructed from the (p−1) variables. Additionally, let $IC(M_{i})$ denote the value of the information criterion associated with the *i*th model $M_{i}$.

Now, consider the following bootstrapping procedure.
(1)Obtain a bootstrap sample denoted as $\left(y^{*},X^{*}\right)$.(2)Fit each of the candidate models to the bootstrap sample.(3)Identify the model $M_{i}$ that corresponds to the minimum information criterion value. In other words, choose model $M_{i}$ such that
$$IC^{*}\left(M_{i}\right)=\min\left(\left\{IC^{*}\left(M_{1}\right),\ldots ,IC^{*}\left(M_{2^{\left(p-1\right)}}\right)\right\}\right),$$
where $IC^{*}\left(M_{i}\right)$ is the bootstrap version of $IC(M_{i})$.(4)Record ${\hat{\beta_{j}}}^{*}$, the maximum likelihood estimate (MLE) of each $\beta_{j}$ corresponding to the covariates represented in the model selected from step 3. The MLEs for unselected variables are set to zero.(5)Repeat the aforementioned steps *B* times.

In our setting, we have that IC=AIC. For the unadjusted results, we let $IC^{*}=AIC^{*}$, where $AIC^{*}=-2\ell \left({\hat{\theta }}^{*}|y^{*}\right)+2k$, ${\hat{\theta }}^{*}={\left[\begin{array}{cc} {\hat{\beta }}^{*T} & {\hat{\sigma }}^{2*} \end{array}\right]}^{T}$, and k=|θ|. On the other hand, for the bias-adjusted results, we have that $AIC^{*}=-2\ell \left({\hat{\theta }}^{*}|y^{*}\right)+3k$.

If we let $y^{*}\left(b\right)$ be a bootstrap sample from *y*, with b∈{1,2,…,B}, and let ${\hat{\beta }}_{j}^{*}\left(b\right)$ be the estimate of the regression coefficient $\beta_{j}$ for the $b^{th}$ bootstrap sample under the model selected for that specific sample, the procedure described above results in a set of estimates given by $\left\{{\hat{\beta }}_{j}^{*}\left(1\right),{\hat{\beta }}_{j}^{*}\left(2\right),\ldots ,{\hat{\beta }}_{j}^{*}\left(B\right)\right\}$. The smoothed estimator of ${\hat{\beta }}_{j}$ is defined as
$${\tilde{\beta }}_{j}^{*}=\frac{1}{B}\sum_{b=1}^{B}{\hat{\beta }}_{j}^{*}\left(b\right).$$

For the confidence interval based on the smoothed estimator, Efron [11] proposes the following approach. Suppose that we have a sample $y={\left[\begin{array}{cccc} y_{1} & y_{2} & \ldots & y_{N} \end{array}\right]}^{T}$, a set of bootstrap samples $\left\{y^{*}\left(1\right),y^{*}\left(2\right),\ldots ,y^{*}\left(B\right)\right\}$ where the elements of this set are of the form $y^{*}\left(b\right)={\left[\begin{array}{cccccc} y_{1}^{*}\left(b\right) & y_{2}^{*}\left(b\right) & \ldots & y_{k}^{*}\left(b\right) & \ldots & y_{N}^{*}\left(b\right) \end{array}\right]}^{T}$, and a bootstrap coefficient estimate ${\hat{\beta }}_{j}^{*}\left(b\right)$, where b={1,2,…,B}. Based on the case when $B=N^{N}$, where $N^{N}$ is equal to all possible bootstrap samples of the form $y^{*}\left(b\right)$, each having a probability of 1/B, if we define
$$Y_{bi}^{*}=\#\left\{y_{k}^{*}\left(b\right)=y_{i}\right\}$$
to be the number of elements of $y^{*}\left(b\right)$ equaling the original data point $y_{i}$, then the smoothed standard error of ${\tilde{\beta }}_{j}^{*}$ can be estimated by
$${\tilde{sd}}_{j}={\left[\sum_{i=1}^{N}{cov}_{ij}^{2}\right]}^{1/2}$$
where
$$cov_{ij}=cov_{*}\left(Y_{bi}^{*},{\hat{\beta }}_{j}^{*}\left(b\right)\right).$$

However, bootstrap applications do not typically consider the “ideal” case with all possible $N^{N}$ bootstrap samples. For the “non-ideal” setting with *B* bootstrap samples and *B* not necessarily equal to $N^{N}$, we can obtain an estimate of the smoothed standard error via
$${\tilde{sd}}_{jB}={\left[\sum_{i=1}^{N}{\hat{cov}}_{ij}^{2}\right]}^{1/2},$$
where
$${\hat{cov}}_{ij}=\sum_{b=1}^{B}\left(Y_{bi}^{*}-Y_{\cdot i}^{*}\right)\left({\hat{\beta }}_{j}^{*}\left(b\right)-{\tilde{\beta }}_{j}^{*}\right)/B.$$

Note that this estimator solely relies on $\left\{y^{*}\left(1\right),y^{*}\left(2\right),\ldots ,y^{*}\left(B\right)\right\}$, which corresponds to the set of bootstrap samples used to estimate ${\tilde{\beta }}_{j}^{*}$. A 95% confidence interval (CI) for the smoothed estimator is called the smoothed 95% CI and it is given by
$${\tilde{\beta }}_{j}^{*}\pm 1.96{\tilde{sd}}_{jB}.$$

## 4. Application in Biomedicine: Sulindac for the Treatment of Colonic and Rectal Ademonams in Patients with Familial Adenomatous Polyposis

Familial adenomatous polyposis (FAP) is an autosomal dominant genetic disease characterized by the development of thousands of polyps throughout the colon and rectum. This condition is rare; it occurs in 1 in 1000 people, and although it accounts for only 1% of all diagnosed colorectal cancers, it is the second most common inherited colorectal cancer syndrome.

FAP is the result of a mutation of a tumor suppressor gene on chromosome 5 known as the ademomatous polyposis coli (APC) gene. Polyps begin to arise in the early teens and if untreated, patients have a 100% lifetime risk of developing colorectal cancer. In addition, patients with FAP are at risk of developing extracolonic pathologies such as desmoid tumors (a solid connective tissue tumor), hepatoblastomas (liver tumors), and thyroid cancer [12].

In patients with FAP, early detection and treatment are essential for preventing the development of colorectal carcinoma, thus improving the prognosis. The standard treatment for FAP is colectomy with or without protectomy. Subtotal colectomy is desirable for many patients, but it requires continued surveillance, while total proctocolectomy does not require surveillance, but patients experience increased stool urgency and higher rates of urinary dysfunction. Therefore, there is a need for the development of non-surgical treatments for patients with FAP [13,14].

In 1993, the effect of Sulindac, a non-steroidal anti-inflammatory drug (NSAID), was investigated for the treatment of FAP in a randomized clinical trial [15]. The study recruited 22 patients with FAP, 11 of whom were assigned to the treatment group that received 150 mg dosages of Sulindac, and 11 of whom were assigned to control group that received an identical-appearing placebo tablet. The study also considered the sex and age of the patients.

The results of the study were published in the paper titled “Treatment of Colonic and Rectal Adenomas with Sulindac in Familial Adenomatous Polyposis” in *The New England Journal of Medicine* [15]. The data set is available in the R package “medicaldata” [16].

The main outcome of interest in the study is the proportionate difference in the number of polyps at 3 months compared to baseline. In other words, if we define *D* as the proportionate change in the number of polyps, then we have that
$$D=\frac{num3pol-baseline}{baseline}.$$

In addition to this outcome, the explanatory variables in the data set include treatment (a value of 1 for Sulindac and 0 for the placebo), sex (a value 1 for male and 0 for female), and age. Moreover, in our modeling analysis, we consider the interaction between treatment and sex, and we define this variable as interaction=treatment·sex.

The primary objective of this application is to assess the selection probability of different linear models that can be constructed with the variables collected from the study. To maintain consistency with the original publication [15], all the candidate models are fitted using least squares regression; an inspection of the distribution of the residuals for the full model confirms that this is a reasonable approach. Ultimately, we will compare the model selection probability estimates that are obtained by utilizing the Akaike weights, the unadjusted bootstrap model frequencies and the bias-adjusted bootstrap model frequencies. The results are displayed in Table 4.

In analyzing the results of this example, several important points emerge. First, it is worth emphasizing that all estimating approaches consistently point towards the model incorporating both the treatment and sex variables as the most likely. This finding is particularly reassuring, as each approach hinges on model selections through AIC.

However, a notable discrepancy arises in the degree to which this favored model is endorsed as reflected by the disparities between the Akaike weights and the BMFs. The Akaike weights suggest a model selection probability of 46.4%, whereas both the adjusted and unadjusted BMFs indicate a higher probability, hovering around 66% each. This distinction carries practical implications, especially in the context of estimating treatment effects and standard errors within the model averaging framework. A higher probability assigned to a single model implies a concentration of estimates from that model, leading to reduced variability due to model selection.

Further scrutiny reveals distinctions between the unadjusted and bias-adjusted BMFs in their estimation of probabilities for the second most popular model. Specifically, for the model containing treatment, sex, and the interaction, the unadjusted BMF yields an estimated probability of 12.3% (in proximity to the Akaike weight for the same model). In contrast, the adjusted version provides a markedly lower estimate of 0.075%. However, for the model involving only treatment, the adjusted approach yields a probability estimate of 16.2%.

To better appreciate the practical effects of these differences in model probability estimation, consider the results on Table 5 and Table 6, which show that the smoothed CIs for the adjusted BMFs are narrower than for the unadjusted case. More importantly, the length of the smoothed CI for treatment is noticeably narrower for the adjusted setting. This occurs because, as noted previously, the second most popular model for the adjusted case is the model that only contains the treatment variable. In other words, 16.2% of the contributions to the smoothed estimate of treatment come from a model that allocates all the data to estimating the effect of treatment. On the other hand, for the unadjusted case, 12.3% of the contributions to the smoothed estimate of treatment come from a model that allocates the data to estimating treatment, sex, and the interaction term.

From an information theoretic point of view, this dissimilarity can be explained by considering the optimism inherent in fitted log-likelihoods. As established in Section 3.2, the fitted log-likelihood given by $-2\ell ({\hat{\theta }}^{*}|y^{*})$ serves as an overly optimistic measure of goodness-of-fit. Thus, to circumvent the unrealistic selection of complex models, the addition of a penalty term of size 2k (as employed in AIC) proves insufficient. Instead, an extra *k* must be incorporated to strike a proper balance between the goodness-of-fit term and the penalty term. Since the unadjusted BMF lacks this extra penalty, it is expected to tend to favor larger models than the adjusted BMF.

## 5. Conclusions

The current practice of statistical modeling recognizes that no single model can capture all of the features of the true data generating model, and that the uncertainty inherent in selecting a model from a candidate collection should ideally be accommodated in the development of inferential procedures. Statisticians have therefore proposed estimating the model parameters of interest by incorporating the contributions of multiple possible candidate models. However, each model’s contribution must be proportional to its probability of being selected.

In this paper, we showed that the AIC-based bootstrap model frequency (BMF) serves as a reasonable approximation to the corresponding AIC-based model selection probability. However, we established that in the process of computing these BMFs, we must add an extra penalty to the bootstrap AIC values of size equal to the number of parameters in the candidate model. This penalty ensures that the bootstrap AIC will remain an asymptotically unbiased estimator of the expected value of the Kullback–Leibler discrepancy. Finally, we showed that the BMFs serve as better approximations to model selection probabilities than the commonly used Akaike weights. We exhibited the superior performance of BMFs through a simulation study, and illustrated their use relative to Akaike weights in a real data application.

## Figures and Tables

**Figure 1 entropy-26-00599-f001:**
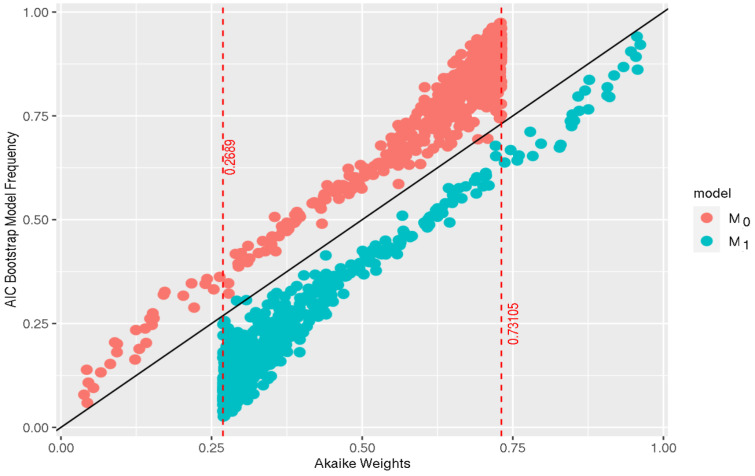
AIC-based bootstrap model frequencies vs. Akaike weights for a null and an alternative model denoted by *M*_0_ and *M*_1_, respectively. The simulation generates 500 data sets from the null model, which is assumed to be true, and each data set is of size *N* = 20. The BMFs are computed with 1000 bootstrap samples.

**Table 1 entropy-26-00599-t001:** Average of the bootstrap model frequencies and the Akaike weights for $M_{0}$ and $M_{1}$ over all 500 simulated data sets.

	AIC-Based BMF	Akaike Weight
*M* _0_	0.7566	0.6166
*M* _1_	0.2434	0.3833

**Table 2 entropy-26-00599-t002:** Results of a simulation with properly specified errors. The simulation consists of 100 data sets from the data generating model, with varying sample sizes. The true model selection probabilities are estimated with the 100 simulated data sets. The BMFs are calculated with 100 bootstraps. The BMFs and Akaike weights are averaged over all 100 data sets.

Sample Size: N = 1000					
*x* _1_	*x* _2_	*x* _3_	BMF	Weights	Probabilities
1	1	1	0.26	0.41	0.23
1	1	0	0.74	0.59	0.77
1	0	1	0.00	0.00	0.00
1	0	0	0.00	0.00	0.00
0	1	1	0.00	0.00	0.00
0	1	0	0.00	0.00	0.00
0	0	1	0.00	0.00	0.00
0	0	0	0.00	0.00	0.00
**Sample Size: N = 100**					
*x* _1_	*x* _2_	*x* _3_	BMF	Weights	Probabilities
1	1	1	0.25	0.38	0.15
1	1	0	0.75	0.62	0.85
1	0	1	0.00	0.00	0.00
1	0	0	0.00	0.00	0.00
0	1	1	0.00	0.00	0.00
0	1	0	0.00	0.00	0.00
0	0	1	0.00	0.00	0.00
0	0	0	0.00	0.00	0.00
**Sample Size: N = 20**					
*x* _1_	*x* _2_	*x* _3_	BMF	Weights	Probabilities
1	1	1	0.28	0.40	0.21
1	1	0	0.69	0.60	0.79
1	0	0	0.02	0.00	0.00
1	0	1	0.01	0.00	0.00
0	1	1	0.00	0.00	0.00
0	1	0	0.00	0.00	0.00
0	0	1	0.00	0.00	0.00
0	0	0	0.00	0.00	0.00

**Table 3 entropy-26-00599-t003:** Results of a simulation with error misspecification. The simulation consists of 100 data sets from the data generating model, with varying sample sizes. The true model selection probabilities are estimated with the 100 simulated data sets. The BMFs are calculated with 100 bootstraps. The BMFs and Akaike weights are averaged over all 100 data sets.

Sample Size: N = 1000					
*x* _1_	*x* _2_	*x* _3_	BMF	Weights	Probabilities
1	1	1	0.24	0.40	0.22
1	1	0	0.76	0.60	0.78
1	0	1	0.00	0.00	0.00
1	0	0	0.00	0.00	0.00
0	1	1	0.00	0.00	0.00
0	1	0	0.00	0.00	0.00
0	0	1	0.00	0.00	0.00
0	0	0	0.00	0.00	0.00
**Sample Size: N = 100**					
*x* _1_	*x* _2_	*x* _3_	BMF	Weights	Probabilities
1	1	1	0.22	0.38	0.18
1	1	0	0.77	0.62	0.82
1	0	1	0.00	0.00	0.00
1	0	0	0.00	0.00	0.00
0	1	1	0.00	0.00	0.00
0	1	0	0.00	0.00	0.00
0	0	1	0.00	0.00	0.00
0	0	0	0.00	0.00	0.00
**Sample Size: N = 20**					
*x* _1_	*x* _2_	*x* _3_	BMF	Weights	Probabilities
1	1	1	0.17	0.40	0.19
1	1	0	0.46	0.60	0.81
1	0	1	0.05	0.00	0.00
1	0	0	0.14	0.00	0.00
0	1	1	0.03	0.00	0.00
0	1	0	0.05	0.00	0.00
0	0	1	0.04	0.00	0.00
0	0	0	0.04	0.00	0.00

**Table 4 entropy-26-00599-t004:** Akaike weights, unadjusted, and adjusted bootstrap model frequencies (BMFs) for models fit to the FAP data set. The BMFs are calculated using 1000 bootstrap samples. Zeros and ones are used to denote the absence or presence of the variable in a selected model. The intercept is included in each model.

Treatment	Sex	Age	Interaction	Weights	Unadjusted	Adjusted
1	1	0	0	**0.464**	**0.656**	**0.668**
1	1	0	1	**0.183**	**0.123**	0.075
1	1	1	0	0.171	0.070	0.050
1	0	0	0	0.073	0.073	**0.162**
1	1	1	1	0.068	0.051	0.021
1	0	1	0	0.039	0.025	0.021
0	1	0	0	0.001	0.000	0.000
0	1	1	0	0.000	0.000	0.001
0	0	1	0	0.000	0.002	0.002
0	0	0	0	0.000	0.000	0.000

**Table 5 entropy-26-00599-t005:** Smoothed estimates and smoothed 95% confidence intervals for regression coefficients from analysis of FAP data set. The results include the bias-adjusted and unadjusted values.

	Adjusted		Unadjusted	
Variable	Smoothed Estimates	Smoothed CIs	Smoothed Estimates	Smoothed CIs
treatment	−0.4609	(−0.6322, −0.2896)	−0.4473	(−0.6567, −0.2379)
sex	0.2498	(0.0122, 0.4874)	0.2773	(0.0101, 0.5444)
age	0.0009	(−0.0050, 0.0068)	0.0012	(−0.0077, 0.0102)
interaction	−0.0205	(−0.1846, 0.1436)	−0.0348	(−0.2824, 0.2127)

**Table 6 entropy-26-00599-t006:** Lengths of smoothed 95% confidence intervals from analysis of FAP data set.

Confidence Interval Lengths		
Variable	Adjusted	Unadjusted
treatment	0.3427	0.4188
sex	0.4752	0.5343
age	0.0118	0.0180
interaction	0.3282	0.4951

## Data Availability

The R code used in generating the data for the simulation study is available on request from the corresponding author. The data for the application are not publicly available since the data set is confidential.

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
