# Peer review of "Bootstrap Approximation of Model Selection Probabilities for Multimodel Inference Frameworks"

_entropy, 2024, doi:10.3390/e26070599_

Round 1

Reviewer 1 Report

Comments and Suggestions for Authors

Comments in an attached file.

Author Response

Comment 1: The scenario considered in the simulation study is very limited. I advise to
consider more covariates (continuous and dichotomous) and other distributions
(for instance, M0: model based on normal distribution and M1: model based
on skew-normal distribution or M0: model based on t-Student distribution and
M1: model based on in the skew-t distribution).

Response: To address this comment, we have included an additional subsection titled ``More Complex Simulation Scenarios," where we exhibit the performance of bootstrap model selection probabilities and Akaike weights in various scenarios, including under distributional misspecification. 

We should emphasize, however, that the performance of AIC under model misspecification falls outside the scope of this paper. Moreover, when we induce distributional misspecification, AIC is not guaranteed to serve as an asymptotically unbiased estimator of the expected KLD. Therefore, the estimated ``true" model probabilities do not necessarily reflect the probabilities that would be governed by the underlying discrepancy targeted by AIC.

Comment 2: I understand that the proposed methodology considers a large sample setting (see Lemma 1, in Section 3). Therefore, I would like to understand why authors consider N=20 in simulation studies? I would like to hear comments from the authors and/or inclusion of simulation studies that investigate the analyzed scenario (and others that will be considered) from which N, the asymptotic
properties are ensured.

Response: We hope that the additional subsection titled ``More Complex Simulation Scenarios" will address both comments 1 and 2, since the added simulation sets consider various sample sizes (N = 20, 100, 1000).   

The simple example provided in the paper considers N=20 because this sample size is sufficiently large for the complexity of the two models at hand. For the larger models in the newly added subsection, N=20 would be considered a small sample size.

Comment 3: It is necessary to make it clear which model is adjusted to the real data. Does D follow a normal distribution? How big is the dataset?
I advise the authors to show the multi-model inference calculations (estimating
the treatment effect and the respective standard errors) so that the impact of
the proposed methodology is clear. From Table 2, I only see that the weights
calculated via different approaches are greater for the same model and even
though for this dataset the adjusted and non-adjusted weights are very close!
Table 2 - In practical terms, which model best fits this set of real data? Just
with treatment, treatment and sex or treatment, sex and interaction? For this
selected model, what does the inference look like? Please show estimates and
standard errors with appropriate interpretations, please.

Response:  The modeling analysis of the Sulindac data set was included to exemplify the theoretical and methodological results established in the paper.  From our perspective, more specific inferential results pertaining to this analysis do not necessarily enhance or alter the significance of the current contribution. 

The clinical trial cited in the paper thoroughly describes the actual effects of Sulindac on the treatment of polyps; hence, our objective is not to further investigate this well-established treatment. In terms of the justification for using least squares regression, we kept the analysis consistent with the original publication. Moreover, we inspected the distribution of residuals and the associated Q-Q plot to make certain the normality assumption was reasonable.

The size and characteristics of the data set had been previously described in the original version of the manuscript: ``The study recruited 22 patients with FAP, 11 of whom were assigned to the treatment group that received 150mg dosages of Sulindac, and 11 of whom were assigned to control group that received an identical-appearing placebo tablet. The study also considered the sex and age of the patients."

The advantages of the multimodel framework over an analysis based on a single model is a widely debated topic in the field of statistics. This paper assumes that the practitioner has already elected to use the multimodel framework. Specifically, we build upon the ideas presented in Efron's 2014 manuscript ``Estimation and Accuracy After Model Selection," published in the Journal of the American Statistical Association. In this paper, Efron argues that the bootstrap can be utilized to account for model selection variability in the estimation of model parameters. In the context of coefficients of linear regression models, this methodology involves averaging coefficient estimates from different models selected through a bootstrap procedure.

The adjusted and unadjusted BMFs are similar for the model with the highest probability, but they differ significantly for the second most probable model. As the reviewer has suggested, the implications of this difference are non-trivial. Therefore, we have added a new subsection to the end of Section 3 to include an overview of how to adapt Efron's multimodel inference framework to estimate regression coefficients and construct associated confidence intervals.  We then utilize this methodology in the FAP application. The results will hopefully clarify the practical importance of employing a biased adjusted BMF.

Because this work is based on the multimodel inference framework, it is not necessarily defensible to designate a ``best" model for the data. In addition, we would like to emphasize that the concept of p-values does not apply either because the framework steers away from assuming that any model is true. Comparing the performance of coefficient estimates, confidence intervals, and evidential measures to assess statistical significance between the multimodel inference framework and the model selection framework is a worthwhile exploration that could serve as the basis for a separate publication.

Comment 4: I would like to hear comments from the authors about the computational cost of
this proposal (please, include some information in the Simulation Section), as in the context of BIG DATA, we have datasets with many explanatory variables,for example, and the authors’ proposal is adjust a collection of models. And
we also have to think about more complex models, where parameter estimation can be computationally intensive. In other words, in what situations would this approach proposed by the authors be most useful in practice?

Response: Big data and high dimensional models indeed present computational challenges in the multimodel framework. For instance, for $p-1$ variables, we must consider $2^{p-1}$ different candidate models. Therefore, if the analysis involves 80 variables, then in principle, we would have to consider $2^{80}$ models, a count that surpasses the estimated number of stars in the observable universe. However, the goal of this paper is not to convince practitioners to use the multimodel framework, though we do personally believe that this is the most defensible approach for addressing the uncertainty that is endemic to the process of statistical modeling. Instead, the goal is to provide a rigorous way to estimate model selection probabilities using likelihood-based information criteria, and to provide guidance when bootstrapping such criteria. Addressing the computational burden in multimodel inference is an open area of statistical research.

Comment 5: The authors have implemented the methods using software R. Is there a new
package? I believe it is important to make the programs available to potential
users. Please, give details about the programming, functions used in R and
make the code available (on Github, for example).

Response: This is an important and valid comment. We indeed recognize the importance of developing and publishing statistical software that practitioners can use to implement new methods. Unfortunately, we have not yet developed an R package for the implementation of this methodology, but we hope to do so in the future.

Reviewer 2 Report

Comments and Suggestions for Authors

The authors have proposed a method to approximate model selection probabilities within multimodel inference frameworks using a bootstrap approach. They have also provided a critique of the Akaike weights as proxies for model probabilities, suggesting a bias-corrected alternative.

Major remarks:

- While the introduction identifies the problem and suggests a general direction for the solution, it could be improved by providing a more explicit statement of the author's contribution. Specifically, it would be beneficial to: (1) Clearly state the main findings or contributions of the study upfront. (2) Provide a succinct overview of the methodology used in the study to set the stage for the detailed explanation that follows in later sections. (3) Briefly mention how their approach compares to or improves upon existing methods.

- The simulation section could benefit from a more detailed description of the methodology to allow for reproducibility. Additionally, simulations are very limited. I propose additional simulations, eg. with more models than 2.

- The comparison between Akaike weights and Bootstrap Model Frequencies (BMFs) should include a more rigorous statistical analysis.

- The article could better highlight the implications of these methods on the practice of statistical modeling in various fields.

- The authors may consider discussing the limitations of their proposed method, such as potential computational complexity or scenarios where the method may not be appropriate.

A real example is interesting, but I would like to see an example where using other than the AIC method of selecting a model could benefit our decision process. In the current example, the selected model is the same.

Minor remarks:

- Vectors and matrices are most commonly bolded in the text.

Author Response

Comment 1:

While the introduction identifies the problem and suggests a general direction for the solution, it could be improved by providing a more explicit statement of the author's contribution. Specifically, it would be beneficial to: (1) Clearly state the main findings or contributions of the study upfront. (2) Provide a succinct overview of the methodology used in the study to set the stage for the detailed explanation that follows in later sections. (3) Briefly mention how their approach compares to or improves upon existing methods.

Response: 

We have added a subsection to the end of Section 3 to provide a succinct yet comprehensive overview of the multimodel framework that serves as the basis for our contributions.  This overview includes a detailed explanation of how to obtain smoothed estimates and smoothed confidence intervals for regression parameters, as proposed by Efron.  We now illustrate this methodology in our application.  

We are uncertain as to how to make our contributions more explicit. In the abstract and the introduction, we clearly articulate that we will demonstrate that estimating model selection probabilities by bootstrapping likelihood-based information criteria induces some bias and that we will present a method for correcting this bias.  Moreover, we show that the bootstrap approach is a more defensible method for estimating model selection probabilities than the widely used Akaike weights.

Comment 2:

 The simulation section could benefit from a more detailed description of the methodology to allow for reproducibility. Additionally, simulations are very limited. I propose additional simulations, eg. with more models than 2.

Response:

To address the limited scope of the simulated example, we have added a new subsection titled ``More Complex Simulation Scenarios".  For all simulation settings, we have provided a detailed description of the generating model and the candidate model collection.

For the application, as previously indicated, we have added a subsection to provide an overview of the multimodel framework proposed by Efron.  We now illustrate this methodology in our application.  

Comment 3:

The comparison between Akaike weights and Bootstrap Model Frequencies (BMFs) should include a more rigorous statistical analysis.

Response:

We are unsure as to what a ``more rigorous statistical analysis" would entail.  Our paper is intended to provide theoretical results to inform important procedural aspects of frequentist-based multimodel inference. The examples and application provided are only meant to illustrate the theory. Nevertheless, we hope that the simulation study in the newly added subsection titled "More Complex Simulation Scenarios" will provide additional clarity.

Comment 4: 

The article could better highlight the implications of these methods on the practice of statistical modeling in various fields.

Response:

Exploring the implications of the multimodel inference framework on statistical practice is indeed a worthwhile topic, but such a discussion is outside of the scope of this contribution. Our paper assumes that the multimodel inference framework is to be adopted, and we provide a frequentist method to estimate model selection probabilities within this framework.

Comment 5:

The authors may consider discussing the limitations of their proposed method, such as potential computational complexity or scenarios where the method may not be appropriate.

Response:

This a valid concern, as addressing the computational limitations of the multimodel inference framework in the presence of big data or high dimensional models is worthwhile area for further research. For instance, for $p-1$ variables, we must consider $2^p-1$ different candidate models. Therefore, if the analysis involves 80 variables, then we have to consider $2^80$ models, a count that surpasses the estimated number of stars in the observable universe. However, the goal of this paper is not to convince practitioners to use the multimodel framework, though we do personally believe that this is the most defensible approach for addressing the uncertainty that is endemic to the process of statistical modeling. Instead, the goal is to provide a rigorous way to estimate model selection probabilities using likelihood-based information criteria, and to provide guidance when bootstrapping such criteria. Addressing the computational burden in multimodel inference is an open area of statistical research, yet one which is outside the scope of our paper.

Comment 6:

A real example is interesting, but I would like to see an example where using other than the AIC method of selecting a model could benefit our decision process. In the current example, the selected model is the same.

Response:

The multimodel framework advocates that practitioners of statistics should avoid selecting a single model and then conducting inference based solely on that model. This perspective aligns with George Box's renowned quote, ``All models are wrong, some are useful." Conventional statistical practice assumes that the selected model represents reality and that no other candidate models were considered.  Such practice ignores the uncertainty that is endemic to the process of statistical modeling. Consequently, the results presented in our contribution are not intended to serve as a new method for selecting a specific model. The model with the highest probability should not be the sole contributor to the estimates of the regression coefficients. Instead, the multimodel approach incorporates all of the models in the candidate collection, with each model's contribution being proportional to its selection probability, which we approximate using the bootstrap.

The objective of this paper is not to persuade the reader to adopt the multimodel framework. The statistical literature already contains numerous papers advocating for the adoption of the multimodel framework over the selection of a single model. Rather, we assume that the practitioner has chosen to adopt this framework, and we provide a frequentist method to estimate model selection probabilities, which are a crucial but not well developed component of this framework.

Round 2

Reviewer 1 Report

Comments and Suggestions for Authors

Accept in present form.

Reviewer 2 Report

Comments and Suggestions for Authors

The responses I have received do not fully address the issues I have raised. However, I accept them because they are quite convincing.